# Higher Periwound Temperature Associated with Wound Healing of Pressure Ulcers Detected by Infrared Thermography

**DOI:** 10.3390/jcm10132883

**Published:** 2021-06-29

**Authors:** Yen-Hsi Lin, Yen-Chin Chen, Kuo-Sheng Cheng, Po-Jui Yu, Jiun-Ling Wang, Nai-Ying Ko

**Affiliations:** 1Department of Nursing, National Cheng Kung University Hospital, Tainan 701, Taiwan; nozomi.yenhsi@gmail.com (Y.-H.L.); yenchin2427@gmail.com (Y.-C.C.); 2Department of Nursing, College of Medicine, National Cheng Kung University, Tainan 701, Taiwan; 3Department of Biomedical Engineering, National Cheng Kung University, Tainan 701, Taiwan; kscheng@mail.ncku.edu.tw; 4Department of Nursing, Fu Jen Catholic University Hospital, Fu Jen Catholic University, New Taipei City 242, Taiwan; pojuiyu@ntu.edu.tw; 5Department of Medicine, College of Medicine, National Cheng Kung University, Tainan 701, Taiwan; 6Department of Internal Medicine, National Cheng Kung University Hospital, Tainan 701, Taiwan

**Keywords:** wound temperature, pressure ulcer, wound healing, pressure ulcer scale for healing, infrared thermography

## Abstract

Visual and empirical assessments do not enable the early detection of wound deterioration or necroses. No suitable objective indicator for predicting poor wound-healing is currently available. We used infrared thermography to determine the association between wound temperature and pressure-wound healing. We examined patients with grades 2–4 pressure ulcers from a medical center in southern Taiwan and recorded the temperatures of the wound bed, periwound, and normal skin using infrared thermographic cameras. A total of 50 pressure ulcers and 248 infrared-thermography temperature records were analyzed. Normal skin temperature was not related to pressure ulcer wound healing. In a multivariate analysis, higher malnutrition universal-screening-tool scores were associated with poor wound-healing (*p* = 0.020), and higher periwound-temperature values were associated with better wound-healing (*p* = 0.028). In patients who had higher periwound-skin temperature than that of the wound bed, that result was also associated with better wound-healing (*p* = 0.002). Wound-bed and periwound temperatures differed significantly with the grade of the pressure ulcer, and a high periwound temperature was positively correlated with wound healing. Infrared thermography can objectively serve as indicators for assessing pressure-ulcer healing.

## 1. Introduction

Pressure-ulcer prevalence is about 0.23% in people aged over 64 years old and from 6.7% to 12.6% in the population receiving home care [1,2]. Visual assessment is the most common method employed by medical personnel for evaluating wound healing among patients with pressure ulcers; however, this method provides limited information on wound necrosis or deterioration and has poor accuracy [3,4]. Assessment tools such as the pressure ulcer scale for healing (PUSH) are limited to the assessment of wound progress [5]; they cannot be used to detect necrosis or infection at an early stage. Advanced imaging technologies such as computed tomography or magnetic resonance imaging are time-consuming, expensive, and involve exposure to contrast media. The judgment of professional radiologists regarding wound treatment is limited by accessibility and timeliness. Therefore, objective, rapid, and easy-to-interpret assessment tools for measuring wound changes are desirable.

Changes in blood flow due to tissue damage and inflammation can influence wound-temperature, and a quantitative measurement of temperature may help the assessment [6,7]. Infrared thermography is a rapid, noncontact, and noninvasive technology [8,9]. The infrared band generated by the thermal radiation of an object is converted into a recognizable temperature pattern by a photosensitive imaging device, which quickly records the radiant energy and temperature distribution released from the human body [10,11,12]. The measurement of test results is applicable in clinical practice for detecting inflammation in subcutaneous tissue or abnormal blood flow. A study reported that infrared thermography facilitated the early detection of changes in wound necrosis or pressure-ulcer progress [13].

A Japanese study explored the correlation between pressure-ulcer temperature and undermining wounds and concluded that pressure-ulcer-induced undermining wounds are common at low temperatures [13]. Farid et al. [14] used infrared thermography to assess the correlation between the temperature of a pressure ulcer and subsequent development of skin necrosis, and they found that the proportion of skin necrosis is significantly higher at low skin temperatures than at high skin temperatures. Cox et al. (2016) found necrosis of deep-tissue injuries is more likely to occur at low skin temperatures [15].

However, no study has focused on the relationship between wound temperature and wound healing related to pressure ulcers. The objective of this study was to investigate the effect of wound temperature measured using infrared thermography on pressure-ulcer wound healing. Moreover, we investigated the association between wound-temperature changes and wound healing.

## 2. Materials and Methods

### 2.1. Study Design

A longitudinal, observational study was conducted to observe changes in pressure-ulcer wounds for 4 consecutive days. This research was approved by the Institutional Review Board of National Cheng Kung University Hospital (NCKUH). This study was conducted in general internal medicine wards in NCKUH, Tainan, Taiwan from 1 August 2019 to 1 August 2020. In the process of data collection, we ensured informed consent and duty of disclosure.

### 2.2. Participants

Patients were included according to the following criteria: (1) Grade 2, Grade 3, or Grade 4 pressure ulcers defined in accordance with the classification standard by the National Pressure Ulcer Advisory Panel (2016) and (2) treatment with wound dressing as instructed. The following patients were excluded: (1) patients scheduled to receive surgery; (2) patients who were hemodynamically unstable or who were dying; (3) patients treated with irremovable wound dressings, such as artificial skin, or negative-pressure wound therapy; (4) patients who experienced rapid progression of wounds, such as obvious wound necrosis or infection, requiring immediate intervention; and (5) patients whose wounds were covered with blisters or scabs.

### 2.3. Instruments (Infrared Thermography)

Infrared thermography: This study employed a FLIR C3 thermal camera as the thermal imaging system. A thermographic camera is a noninvasive, noncontact handheld infrared-detection device with an infrared-detection resolution of 80 × 60 pixels, a visible-light resolution of 640 × 480 pixels, a temperature range of −10 °C to 150 °C, accuracy of ±2% (±3.6 °F), and thermal sensitivity of <0.1 °C. The device had received the following certifications: International Organization for Standardization (ISO)/TR 13154:2009; CE/FCC, CEC, BC, EN61233; and Waste Electrical and Electronic Equipment 2012/19; RoHs 2011/65/ec; C-Tick; EN61000-6-3 of the International Electrotechnical Commission.

The FLIR C3 infrared thermal camera is a thermal-imaging system comprising a noninvasive, noncontact handheld infrared-detection device with an infrared-detection resolution of 80 × 60 pixels, a visible-light resolution of 640 × 480 pixels, a temperature range of −10 °C to 150 °C, accuracy of ±2%, and thermal sensitivity of <0.1 °C. Figure 1 presents the adopted thermal imaging method.

Five infrared thermal images were obtained at each dressing change. High-resolution images were selected to record temperature differences. Specifically, infrared thermal images represent the temperature of the wound bed, periwound skin, and normal skin surrounding the pressure-ulcer wound. Because the wound bed is at the center of the wound, its temperature is referred to as the core temperature of the wound (IRTc) in infrared thermography. The temperature of the periwound skin—the area surrounding the wound’s edge—is denoted by IRTp in infrared thermography. The temperature measured in normal skin (3–5 cm from the wound) is defined as the skin temperature around the wound and denoted by IRTn in infrared thermography.

### 2.4. Wound Assessment

According to different pressure-ulcer temperatures in various areas, the patients were classified into two groups: the high-temperature group, in which the temperature of the periwound skin was higher than that of the wound bed (IRTp > IRTc), and the low-temperature group, in which the temperature of the periwound skin was lower than that of the wound bed (IRTp < IRTc).

We applied the PUSH Tool 3.0 to conduct clinical assessment of wound healing progression [16,17]. We compared the total PUSH score in the subsequent 1 and 3 days with the baseline. We subtracted the score on Day 0 from the scores on Day 1 and Day 3. Higher scores indicate worse ulcer-healing conditions, and patients with higher scores were classified as the poor wound-healing group. Diminishing scores indicate improvement in the wound healing process, and patients with lower scores were classified as the good wound-healing group.

### 2.5. Data Collection

Wound photography of the pressure ulcers was performed before wound cleaning. To maintain the credibility and reliability of data, researchers took the photograph at the same time of day for each patient and placed the patient in the same position during wound dressing changes. After removing all wound dressings, the researcher rinsed and thoroughly dried the wounds. The wound area remained free of pressure for 10–15 min. To accurately record wound changes, at least five images were captured each time. Patients’ demographic characteristics, including age, body mass index (BMI), laboratory data (hemoglobin (Hb), albumin), comorbidities, and pressure-ulcer characteristics (PUSH, Braden scale) were collected from the patients’ medical records.

### 2.6. Statistical Analysis

Descriptive statistics are presented as means and standard deviations. The chi-square test was performed to determine the correlation between pressure-ulcer wound temperatures and the wound healing process. A generalized estimating equation was applied to explore factors affecting the changes in pressure-ulcer wounds. All statistical analyses were performed using SPSS version 17.0.

## 3. Results

### 3.1. Demographic Characteristics and Pressure Ulcer Characteristics

In the study, data on 50 pressure ulcers were collected from 37 individuals. The observation period was from 3 to 10 days for each wound (mean, 4.18 days). A total of 248 infrared temperature measurements were obtained.

The participants in the experiment were aged 40–88 years, with a mean age of 76.62 years (standard deviation [SD] = 9.37), and 88% were older than 65 years (Table 1). The mean body mass index (BMI) of participants was 18.75 (SD = 4.54), and 48.6% (*n* = 18) of individuals were categorized as underweight. A total of 56.7% had a malnutrition universal screening tool (MUST) score of >2. The mean hemoglobin (hb) and albumin levels were 9.71 g/dL (SD = 1.63) and 2.62 g/dL (SD = 0.54), respectively.

Pressure-ulcer wounds were located at the sacrum/coccyx, hip bone, rib, scapula, ischium, back, ilium, and ankles (Table 2). Pressure ulcers were observed in the sacrococcygeal bone (56%), ischium (12%), and back (10%). The largest portion of pressure ulcers were classified as grade 3 ulcers (40%). The initial mean PUSH score was 12.72 (SD = 3.10), with most participants (56%) having a score of 10–14. The mean score of the Braden Scale for Predicting Pressure Ulcer Risk was 11.1 points (SD = 2.33).

We recorded the healing process of 248 wounds, and participants were categorized into good and poor wound-healing groups according to their PUSH score change on the subsequent 1 and 3 days. As shown in the scatter plots (Figure 2), the periwound skin temperature was lower than the temperatures of the wound bed and normal skin in the poor wound-healing group, and the wound-bed temperature gradually decreased as the wound healed. In the good wound-healing group, the periwound skin temperature was higher than the temperatures of the wound bed and normal skin. Appendix A showed case demonstrations with temperature and wound condition.

### 3.2. Factors Influencing the Healing of Pressure Ulcer Wounds

A generalized estimating equation was used to analyze the influence of all factors on the wound-healing process (good or poor wound-healing). Univariate and multivariate analyses demonstrated that patients’ MUST score was related to their wound healing (Table 3). In the multivariate analysis, higher MUST scores were associated with poor wound-healing (*p* = 0.020), and higher IRTp values were associated with better wound healing (*p* = 0.028). The high-temperature group was also associated with better wound healing (*p* = 0.002).

When we changed the frequency of observation to once every three days (Table 4), the nutritional status (MUST scores) and temperature of the periwound skin (IRTp) remained correlated with wound-healing.

The examples of good and poor wound-healing are shown in Figure 3 and Figure 4.

## 4. Discussion

We conducted analysis to determine if the wound bed, periwound skin, and normal skin temperatures, measured by infrared thermometer, were correlated with wound healing. We found that high periwound skin temperature was associated with good wound-healing compared with low periwound skin temperature.

Goto et al. [18] and Kanazawa et al. [13] applied infrared thermography for evaluating normal wound-healing, and they discovered the highest temperature in the periwound skin, with a lower temperature in the normal skin surrounding the wound and the lowest temperature in the wound bed. Our results concur with the findings of Kanazawa et al. [13] in that a lower periwound skin temperature was associated with an increased risk of latent necrosis. Nakagami et al. [19] discovered that delayed wound healing is likely to occur in people whose wound-bed temperature is higher than their periwound skin temperature. This can be explained by the physiological mechanism that body heat generated from cell metabolism spreads across the body through blood flow and increases blood perfusion [15].

In this study, factors found to influence wound healing included nutritional status and changes in periwound skin temperature. Stress injury is related to nutritional status. Malnourished people exhibit weaker immunity, increased risks of infection, and delayed wound healing, as indicated by their reduced BMI, hemoglobin, and albumin index values [20,21]. Obese and underweight people are high-risk groups for pressure ulcers [22].

Literature has shown that a rise in wound temperature may imply inflammation, immune responses, vasodilation induced by inflammatory factors, and tissue metabolism [13,19]. Wound temperature may be related to proximity to blood flow or compromised blood flow and deep acute inflammatory conditions [23]. Elevated venous ulcer temperature indicates skin inflammation [18]. Nishide [24] tested the inflammatory response of diabetic feet and identified wound temperature measurement to be an indicator of wound recovery. We noted changes in the temperatures of the wound bed, periwound skin, and normal skin during wound healing. In addition, better wound-healing occurred when the temperature of the periwound skin was higher than that of the wound bed. Such temperature changes might be attributable to the re-epithelialization at the edge of a full-thickness wound during the healing process [25]. In our study, patients with overt infection were excluded. Except slough or eschar, the skin temperatures of wound bed or periwound skin were usually elevated in infected wound [26]. Some examples of infected wounds are shown in Appendix A. However, some experts suggest looking for more clinical signs of tissue infection when an increased local temperature is detected [23].

This research has the following limitations: (1) The short hospitalization times of patients provided a mean observation period of only 4.5 days. This study thus failed to conduct long-term observations of the healing process until wound closure. (2) The infrared instrument had high resolution and sensitivity, but the equipment was slightly bulky and heavy overall. Operation of the equipment thus requires improvement. (3) The blood-flow distribution and temperature differ in different parts of the body (e.g., trunk and limb), and this might have affected the pressure-ulcer wound-evaluation results. (4) Infrared thermography is limited to the monitoring of fully open wounds. Deep-tissue injuries and unstageable pressure ulcers, covered by eschar and slough, could not be fully examined and may require the use of 3-D ultrasound for assessment. (5) The sample in this study relates to patients with pressure ulcers only and may not be more widely generalizable to other wound.

## 5. Conclusions

Evaluating pressure ulcers by infrared thermography can help predict prognosis better than solely using visual assessment. Temperature variations can be used to determine wound size. When lower temperatures of the periwound skin than of the wound bed were detected by infrared thermography, poor wound-healing is possible, and appropriate interventions are necessary.

## Figures and Tables

**Figure 1 jcm-10-02883-f001:**
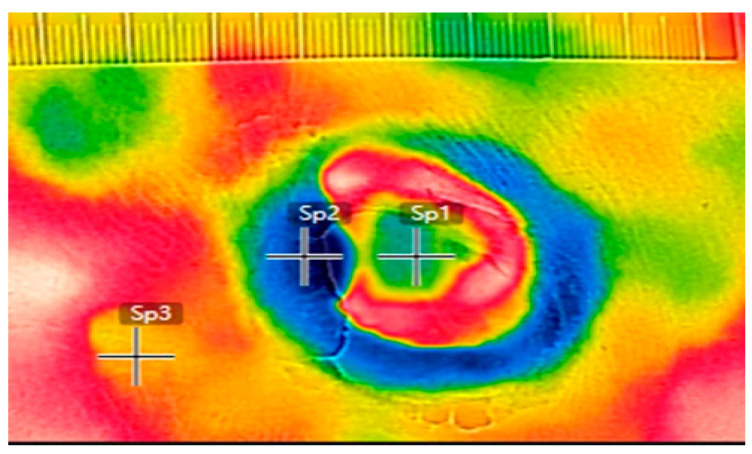
Infrared thermal imagery of a wound. Sp1: area of wound bed; Sp2: area of periwound skin; Sp3: area of normal skin.

**Figure 2 jcm-10-02883-f002:**
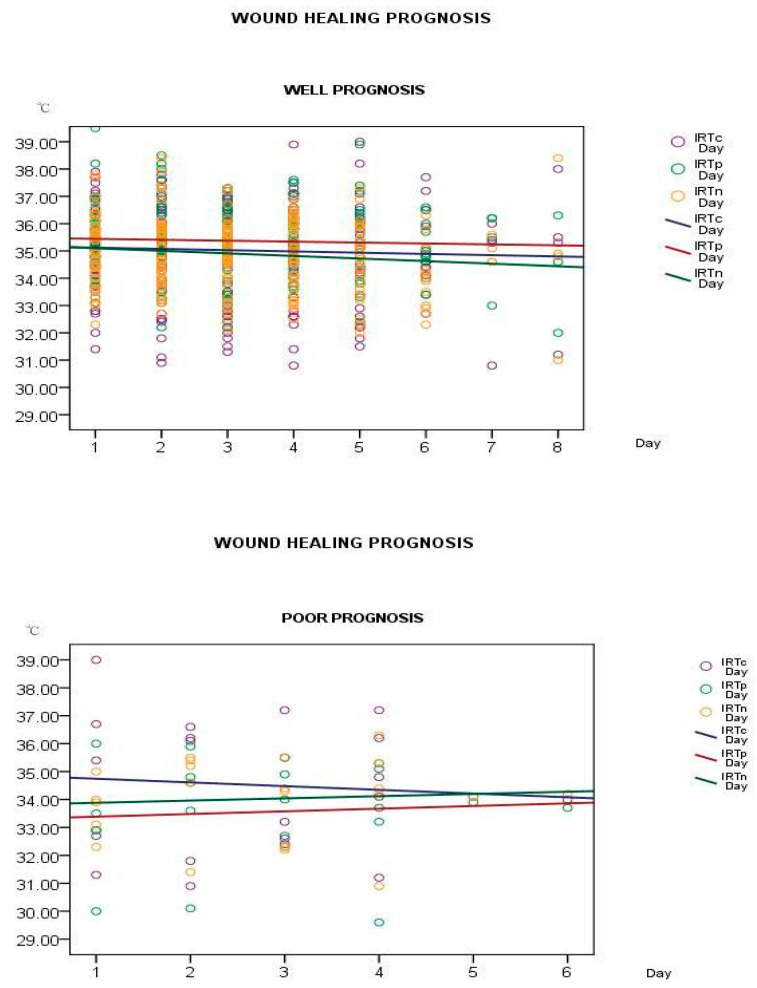
Changes in the temperature of pressure wounds. IRTc, wound-bed temperature; IRTp, periwound skin temperature; IRTn, normal skin temperature. The upper picture showed the temperature change in better wound healing. Periwound skin temperatures (red line) were higher than wound-bed temperatures (blue line) and normal skin temperatures (green line). The mean periwound skin temperature was more than 35 °C. The lower picture showed the temperature change in poor wound-healing. Periwound skin temperatures (red line) were lower than wound-bed temperatures (blue line) and normal skin temperatures (green line). The mean periwound skin temperature was less than 34 °C.

**Figure 3 jcm-10-02883-f003:**
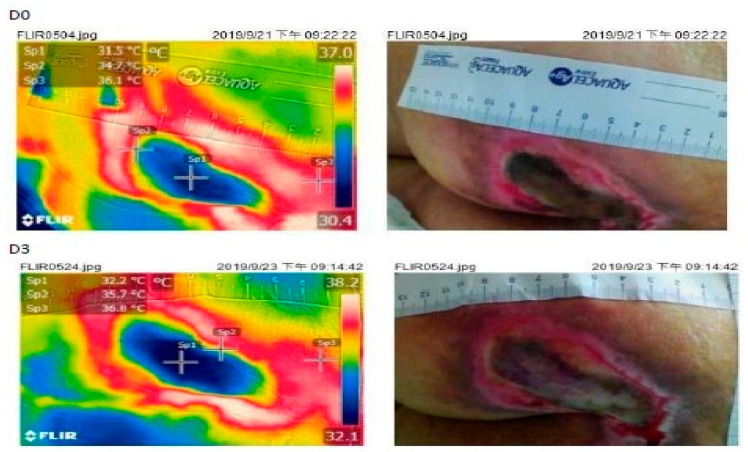
Case of better wound healing (upper: Day 0 and lower: Day 3). IRTc: wound bed (Sp1) IRTp: periwound skin (Sp2); IRTn: normal skin (Sp3).

**Figure 4 jcm-10-02883-f004:**
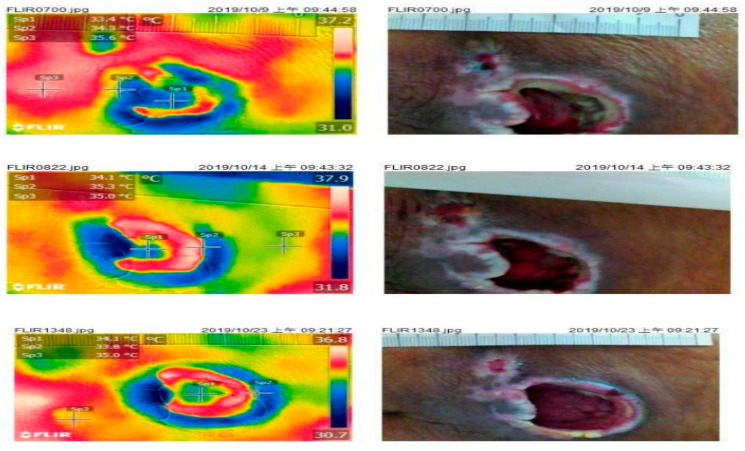
Case of poor wound-healing (upper to lower in the consecutive days). IRTc: wound bed (Sp1) IRTp: periwound skin (Sp2); IRTn: normal skin (Sp3).

**Table 1 jcm-10-02883-t001:** Demographic data and clinical characteristics (*n* = 37).

Variable	*n* (%)	Mean ± SD	Range
Age		76.62 (±9.37)	40–88
<65	5 (13.5%)		
65–69	5 (13.5%)		
70–74	4 (10.8%)		
75–79	8 (21.6%)		
80–84	5 (13.5%)		
>85	10 (27%)		
Sex			
Men	16 (43.2%)		
Women	21 (56.8%)		
Nutrition			
BMI		18.75 (±4.54)	9.57–29.38
<18.5	18 (48.6%)		
18.5–24	15 (40.4%)		
24–27	2 (5.4%)		
>27–30	2 (5.4%)		
MUST		1.68 (±1.20)	
0	7 (18.9%)		
1	9 (24.3%)		
2	14 (37.8%)		
3	3 (8.1%)		
4	4 (10.8%)		
Hb		9.71 (±1.63)	
Albumin		2.62 (±0.54)	
Comorbidity			
CCI		13.14 (±2.87)	
Diabetes mellitus			
Yes	23 (62.1%)		
No	14 (37.8%)		
HbA1C	12	6.3%	

BMI: body mass index; CCI: Charlson comorbidity index; MUST: malnutrition universal screening tool; Hb: hemoglobin.

**Table 2 jcm-10-02883-t002:** Pressure-wound characteristics (*n* = 50).

Variable	*n* (%)	Mean ± SD
Wound site		
Sacrococcygeal	28 (56%)	
Ischium	6 (12%)	
Back	5 (10%)	
Hip bone	4 (8%)	
Ilium	3 (6%)	
Scapula	2 (4%)	
Rib	1 (2%)	
Ankle	1 (2%)	
Grade of pressure ulcer		
Grade. 2	18 (36%)	
3	20 (40%)	
4	12 (24%)	
PUSH score		12.72 (3.10)
<10	8 (16%)	
10–14	28 (56%)	
>15	14 (28%)	
Braden Scale		11.1 (2.33)
<10	23 (46%)	
11–15	25 (50%)	
>15	2 (4%)	

**Table 3 jcm-10-02883-t003:** Factors associated with poor wound-healing (increased PUSH score) after 1 day (*n* = 248).

Variable	Univariate Analysis		Multivariate Analysis	
	ß (95% Confidence Interval)	*p*	ß (95% Confidence Interval)	*p*
Age	1.025 (0.966, 1.087)	0.421		
BMI	0.740 (0.530, 1.034)	0.077		
MUST	3.445 (1.187, 9.999)	0.023 **	3.354 (1.209, 9.307)	0.020 **
CCI	1.042 (0.587, 1.848)	0.888		
Hb	1.151 (0.629, 2.107)	0.647		
Albumin	1.242 (0.073, 21.10)	0.881		
IRTc	0.866 (0.508, 1.420)	0.568		
IRTp	0.420 (0.260, 0.679)	<0.001 **	0.469 (0.239, 0.921)	0.028 *
IRTn	0.610 (0.358, 1.039)	0.069		
High-temperature group	0.206 (0.076, 0.556)	0.002 **	0.200 (0.074, 0.546)	0.002 **
Low-temperature group	(Reference value)			

* *p* < 0.05; ** *p* < 0.01. BMI: body mass index; CCI: Charlson comorbidity index; MUST: malnutrition universal screening tool; Hb: hemoglobin; IRTc, wound-bed temperature; IRTp, periwound skin temperature; IRTn, normal skin temperature. High-temperature group: periwound skin temperature > wound-bed temperature (IRTp > IRTc); low-temperature group: periwound skin temperature < wound-bed temperature (IRTp < IRTc).

**Table 4 jcm-10-02883-t004:** Factors associated with poor wound-healing (increased PUSH score) after three days (*n* = 248).

Variable	Univariate Analysis		Multivariate Analysis	
	ß (95% Confidence Interval)	*p*	ß (95% Confidence Interval)	*p*
Age	0.974 (0.915, 1.036)	0.394		
BMI	1.330 (0.954, 1.852)	0.092		
MUST	0.304 (0.110, 0.838)	0.021 **	0.311 (0.119, 0.922)	0.034 *
CCI	0.884 (0.522, 1.496)	0.646		
Hb	0.961 (0.543, 1.700)	0.891		
Albumin	1.359 (0.078, 23.766)	0.833		
IRTc	1.068 (0.575, 1.983)	0.835		
IRTp	3.161 (1.235, 8.091)	0.016 **	3.108 (1.193, 8.095)	0.020 *
IRTn	1.967 (1.014, 3.816)	0.066	0.682 (0.373, 1.245)	0.212
High-temperature group	5.684 (1.378, 23.448)	0.016 **	4.862 (1.797, 13.151)	0.002 **
Low-temperature group	(Reference value)			

* *p* < 0.05; ** *p* < 0.01. BMI: body mass index; CCI: Charlson comorbidity index; MUST: malnutrition universal screening tool; Hb: hemoglobin: Hb;IRTc, wound-bed temperature; IRTp, periwound skin temperature; IRTn, normal skin temperature. High-temperature group: periwound skin temperature > wound-bed temperature (IRTp > IRTc); low-temperature group: periwound skin temperature < wound-bed temperature (IRTp < IRTc).

## Data Availability

The data can be made available after contacting the corresponding author.

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
