# Peer review of "Higher Periwound Temperature Associated with Wound Healing of Pressure Ulcers Detected by Infrared Thermography"

_jcm, 2021, doi:10.3390/jcm10132883_

Round 1

Reviewer 1 Report

The manuscript reports on a clinical study where wound temperature measured by infrared thermography was tested as a new method for the evaluation of pressure wound healing. The reported results are of relevance as there is a need of new tools for the clinicla evaluation of wound status in order to chose the most appropriate treatment. 

There are a few points that need to be improved:

  1. Figure 2 needs to be improved. The legend is not visible. It is not possible to understand the shown data.
  2. Discussion. Factors as infections or blood perfusion can influence tissue temperature and results should be discussed also from this point of view. Were patient data on infection or impared perfusion available? 
  3. Conclusions need to be improved. 

Author Response

1.Figure 2 needs to be improved. The legend is not visible. It is not possible to understand the shown data.

Ans: We have added the legend. The upper picture showed the temperature change of better wound healing. Periwound skin temperature (red line) were higher than wound bed temperature (blue line) and normal skin temperature (green line). The mean periwound skin temperature was more than 35C. The lower picture showed the temperature change of poor wound healing. Periwound skin temperature (red line) were lower than wound bed temperature (blue line) and normal skin temperature (green line). The mean periwound skin temperature was less than 34C

2 Discussion. Factors as infections or blood perfusion can influence tissue temperature and results should be discussed also from this point of view. Were patient data on infection or impared perfusion available? 

Ans: We have add some discussion about infection and blood perfusion.

The wound temperature maybe related to proximity to blood flow, or compromised blood flow and deep acute inflammatory conditions.

 In our study, patients with overt infection were excluded. Except slough or eschar, the skin temperatures of wound bed or periwound skin were usually elevated in infected wound[26]. Some examples of infected wound were shown in Supplementary materials ( Figure S2). However , some experts suggest look for more clinical signs of tissue infection when an increased local temperature is detected[23].

3.Conclusions need to be improved. 

Ans: We have revised as the following. Evaluating pressure ulcer by infrared thermography can help predict prognosis better than by only visual assessment. Temperature variations can be used to determine wound size.  When lower temperature of the periwound skin than that of the wound bed was detected by infrared thermography,  a poor wound healing is possible and appropriate interventions is necessary.

Reviewer 2 Report

This is an interesting study of a very current topic. There is a new paper that's just been published that I think you should consider referring to:

Gethin, G, Ivory, JD, Sezgin, D, Muller, H, O'Connor, G, Vellinga, A. What is the “normal” wound bed temperature? A scoping review and new hypothesis. Wound Rep Reg. 2021; 1– 5.

Title of study: sentences 2 and 3. I would advise that the title include pressure ulcer or pressure injury as this will help indexing and retrieval. 

Regarding the terms pressure ulcer or pressure injury you need to decide which term to use and be consistent throughout the paper.

Sentence 25 - pressure ulcer (or injury) should be plural i.e. 50 pressure ulcers or 50 pressure injuries

Sentence 39: Pressure ulcer (or injury) is......remove The at the beginning of the sentence

When you decide on the term pressure ulcer or injury make sure the plural / single form is used correctly throughout.

Sentence 44: reference(s) needed for the PUSH tool

Sentence 54/55 Grammar of sentence needs reviewing

Sentence 85: suggest changing unremovable to irremovable

Sentence 88: we would more commonly refer to scabs rather than crusts

Sentence 91: A thermographic camera.....(remove the word 'said')

Sentence 106: Suggest sentence reads -..........obtained at each dressing change.....

Sentence 130: Suggest.......the research took the photo at the same time of day for each patient.....

Sentence 158: suggest replacing the term 'tailbone' for the anatmocially correct term

Sentence 233 - suggest Limitations are presented as a separate subsection. Also in this section include a comment that the sample relates to patients with pressure ulcers / injuries only so may not be more widely generalizable.

Sentence 244: Sentence needs editing as it does not make sense as written.

Conclusion - needs to be stronger and reflect the specific temperature thresholds, also consider what further research is needed and what are the clinical implications of the findings?

Author Response

Gethin, G, Ivory, JD, Sezgin, D, Muller, H, O'Connor, G, Vellinga, A. What is the “normal” wound bed temperature? A scoping review and new hypothesis. Wound Rep Reg. 2021; 1– 5.

Ans: We have added this reference.(now in Ref.6)

Title of study: sentences 2 and 3. I would advise that the title include pressure ulcer or pressure injury as this will help indexing and retrieval.

Ans: We have revised.

Regarding the terms pressure ulcer or pressure injury you need to decide which term to use and be consistent throughout the paper.

Ans: We have revised.

Sentence 25 - pressure ulcer (or injury) should be plural i.e. 50 pressure ulcers or 50 pressure injuries

Ans: We have revised.

Sentence 39: Pressure ulcer (or injury) is......remove The at the beginning of the sentence

Ans: We have revised.

When you decide on the term pressure ulcer or injury make sure the plural / single form is used correctly throughout.

Ans: We have revised.

Sentence 44: reference(s) needed for the PUSH tool

Ans: We have revised. (now Ref.5)

Sentence 54/55 Grammar of sentence needs reviewing

Ans: We have revised.

Sentence 85: suggest changing unremovable to irremovable

Ans: We have revised.

Sentence 88: we would more commonly refer to scabs rather than crusts

Ans: We have revised.

Sentence 91: A thermographic camera.....(remove the word 'said')

Ans: We have revised.

Sentence 106: Suggest sentence reads -..........obtained at each dressing change.....

Ans: We have revised.

Sentence 130: Suggest.......the research took the photo at the same time of day for each patient.....

Ans: We have revised.

Sentence 158: suggest replacing the term 'tailbone' for the anatmocially correct term

Ans: We have revised.

Sentence 233 - suggest Limitations are presented as a separate subsection. Also in this section include a comment that the sample relates to patients with pressure ulcers / injuries only so may not be more widely generalizable.

Ans: We have revised.

Sentence 244: Sentence needs editing as it does not make sense as written.

Ans: We have revised.

Conclusion - needs to be stronger and reflect the specific temperature thresholds, also consider what further research is needed and what are the clinical implications of the findings?

Ans: We have revised. “Evaluating pressure ulcer by infrared thermography can help predict prognosis better than by only visual assessment. Temperature variations can be used to determine wound size. When lower temperature of the periwound skin than that of the wound bed was detected by infrared thermography, a poor wound healing is possible and appropriate interventions is necessary.”